# CD123 as a Therapeutic Target in the Treatment of Hematological Malignancies

**DOI:** 10.3390/cancers11091358

**Published:** 2019-09-12

**Authors:** Ugo Testa, Elvira Pelosi, Germana Castelli

**Affiliations:** Department of Oncology, Istituto Superiore di Sanità, Viale Regina Elena 299, 00161 Rome, Italy; elvira.pelosi@iss.it (E.P.); germana.castelli@iss.it (G.C.)

**Keywords:** leukemia, acute myeloid leukemia, interleukin-3, interleukin-3 receptor, leukemic stem cells, therapy, biomarker

## Abstract

The interleukin-3 receptor alpha chain (IL-3Rα), more commonly referred to as CD123, is widely overexpressed in various hematological malignancies, including acute myeloid leukemia (AML), B-cell acute lymphoblastic leukemia, hairy cell leukemia, Hodgkin lymphoma and particularly, blastic plasmacytoid dendritic neoplasm (BPDCN). Importantly, CD123 is expressed at both the level of leukemic stem cells (LSCs) and more differentiated leukemic blasts, which makes CD123 an attractive therapeutic target. Various agents have been developed as drugs able to target CD123 on malignant leukemic cells and on the normal counterpart. Tagraxofusp (SL401, Stemline Therapeutics), a recombinant protein composed of a truncated diphtheria toxin payload fused to IL-3, was approved for use in patients with BPDCN in December of 2018 and showed some clinical activity in AML. Different monoclonal antibodies directed against CD123 are under evaluation as antileukemic drugs, showing promising results either for the treatment of AML minimal residual disease or of relapsing/refractory AML or BPDCN. Finally, recent studies are exploring T cell expressing CD123 chimeric antigen receptor-modified T-cells (CAR T) as a new immunotherapy for the treatment of refractory/relapsing AML and BPDCN. In December of 2018, MB-102 CD123 CAR T developed by Mustang Bio Inc. received the Orphan Drug Designation for the treatment of BPDCN. In conclusion, these recent studies strongly support CD123 as an important therapeutic target for the treatment of BPDCN, while a possible in the treatment of AML and other hematological malignancies will have to be evaluated by in the ongoing clinical studies.

## 1. Introduction

### 1.1. Acute Myeloid Leukemia

Currently, there are three different approaches to try to improve the treatment of acute leukemia: (i) The targeting of a molecular abnormality specific of a leukemia subset using highly specific drugs; (ii) the targeting of a membrane protein selectively or preferentially expressed on leukemic cells, compared to the normal counterpart, usually using a specific monoclonal antibody, conjugated or not with a cytotoxic drug; (iii) the targeting of a molecule not specifically expressed in leukemic cells, but essential for leukemic cells, while sparing normal hematopoietic stem cells; (iv) targeting membrane proteins specifically or preferentially expressed on leukemic cells with specific chimeric antigen receptor-modified T-cells (CAR T) and (v) a new type of anti-leukemic drugs that, at variance with classical anti-leukemia chemotherapeutic agents do not exert a cytotoxic effect on leukemic cells, but act as epigenetic modifiers of the gene expression profile of leukemic cells.

Acute myeloid leukemia (AML) and myelodysplastic syndromes (MDS) are heterogeneous disorders originated from hematopoietic stem cells (HSCs) through the progressive and sequential acquisition of genetic and epigenetic alterations that cause a clonal expansion of myeloid progenitors/precursors in the bone marrow and peripheral blood, exhibiting disease-specific phenotypes mainly represented by impaired cell differentiation and increased proliferation [1]. AML is actually cured in about 35–40% of patients with <60 years but the prognosis for AML patients >60 years is improving but remains poor [1]. 

Studies carried out during the last decade have provided evidence that AML derives from a series of genetic alterations accumulated at the level of HSCs with age. These studies have provided evidence that clonal hematopoiesis (CH), characterized by the expansion of HSCs and hematopoietic progenitor cells (HPCs) that carry somatic mutations but are still capable of normal cell differentiation, can precede the occurrence of overt leukemia by many years [2]. These clonal neoplasms may transform to overt leukemia over time: Depending on the number and type of somatic CH mutations can be subdivided into CH with indeterminate potential (CHIP) and CH with oncogenic potential (CHOP) [3]. The consistent difference between these two conditions is that, while CHIP mutation provides the molecular background of a leukemic process, CHOP mutations are in part leukemia-specific mutations that affect cell differentiation and proliferation of hematopoietic cells [3]. Thus, the passenger mutations observed in CHIP are mainly represented by mutations at low variant allelic frequency at the level of genes such as *TET2, DNMT3A, GNAS, ASXL1, SF3B1, PPM1D* and per se are not sufficient to cause the development of a leukemic process [3]. In contrast, CHOP-specific mutations are represented by driver mutations occurring at the level of genes such as *BCR-ABL1, JAK2, RUNX, KRAS* and *HRAS* at higher variant allelic frequency [3].

AMLs are characterized by a consistent genetic heterogeneity; genetic alterations are recurrent and include amplifications, deletions, rearrangements and point mutations. AMLs have been classified according to their origin, morphology, cytogenetic and molecular aberrations. Concerning the origin, AMLs are classified into: (i) De novo AML; (ii) therapy-related AML (t-AML), associated with prior chemotherapy with potentially mutagenic drugs and (iii) secondary AML (s-AML) associated with a prior myelodysplastic syndrome or a myeloproliferative disorder [4]. Prognostic risk of AMLs is defined at diagnosis according to the presence of specific cytogenetic and molecular aberrations [5,6,7]. Criteria for AML classification and risk stratification have been proposed by several organizations, including the European Leukemia NET (ELN) [5], National Comprehensive Cancer Network (NCCN) [6] and World Health Organization (WHO) [7]. The NCCN and ELN guidelines stratify AML patients into three different risk groups: Favorable, intermediate and poor/adverse [5,6]. The most adopted risk classification is the ELN risk stratification: Patients are classified into one of the four risk groups, including favorable, intermediate 1, intermediate 2 and adverse (Table 1). Favorable prognosis group includes AMLs with acute promyelocytic leukemia (APL) t(15;17)(q22;q12), balanced translocations t(8;21)(q22;q22), biallelic mutated CBPA and inv(16)(p13.1q22), mutated *NPM1* without *FLT3-ITD* or with *FLT3-ITD*^low^. Intermediate 1 group comprises mutated *NPM1* with *FLT3-ITD*, WT*-NPM1* with or without *FLT3-ITD*. The intermediate 2 group comprises t(9;11), *MLLT3-MLL* and cytogenetic abnormalities neither favorable or adverse. The adverse AML group comprises AMLs with complex karyotype, inv(3)(q21q26)/t(3;3)(q21;q26), *DEK-NUP214* t(6;9)(p23;q34), *RPN1-EVI1*, t(6;11), −5 or del(5q), −7 or abnormal (17p) or monosomal karyotype. NCCN and ENL adopt a similar classification scheme for favorable-risk AMLs, although the criteria for favorable risk differ in some respects in these two evaluation systems [5,6]. It is important to point out that each of these risk groups is relatively heterogeneous, even considering the favorable-risk AML group. Thus, a consistent genotypic and clinical heterogeneity exists within the favorable risk AML.

The advent of next generation sequencing has allowed to define the genetic landscape of AML, with the discovery of some recurrent genetic alterations, some are “driver” mutations and play a key role in leukemic development, while other ones are “passenger” mutations and do not play a major role in leukemia development and just reflect the genomic instability of these tumor cells. The most frequent, recurring mutations are represented by: (i) Nucleophosmin 1 (*NMP1*) mutations, occurring in 25–30% of AML patients and, in the absence of *FLT3-ITD* mutations, predict favorable overall survival; (ii) DNA methyltransferase 3A (*DNMT3A*) mutations, occurring in about 20–25% of all AMLs and representing mutations observed in pre-leukemic lesions, originating early during leukemia development and persisting during disease remission; (iii) Fms-like tyrosine kinase 3 (*FLT3*) mutations, consisting in either by internal tandem duplications (ITD) in the juxta-membrane domain or in mutations in the second tyrosine kinase domain (TKD) of the *FLT3* gene and have been observed in 20–25% of all AMLs, particularly those with normal karyotype: *FLT3-ITD*, particularly when the mutant allele is well expressed, is associated with poor prognosis; (iv) isocitrate dehydrogenase (*IDH*) mutations occur in about 15–20% of all AMLs, at the level of either the IDH1 or IDH2 gene and are gain-of-function mutations conferring the loss of the physiologic enzyme function and the acquisition of a new enzymatic function, consisting in the conversion of α-ketoglutarate into 2-hydroxyglutarate and are usually associated with lower overall survival; (v) ten-eleven translocation 2 (*TET2*) mutations, observed in 15–20% of cases, are associated with clonal hematopoiesis in healthy elderly subjects; (vi) runt-related transcription factor (*RUNX1*) mutations are observed in 5–15% of AMLs, associated with trisomy 13, trisomy 21, absence of *NPM1* mutations: This mutation makes part of AMLs with mutated chromatin, RNA splicing or both and is frequent in s-AMLs; (vii) *NRAS* signal transducer is mutated in about 15–20% of AMLs and is associated with the NPM1 and biallelic CCAAT enhancer binding protein α (CEBPA) mutation; (viii) CCAAT enhancer binding protein α (*CEBPA*) mutations are observed in 5–10% of all AMLs, frequently in association with del (9q): 4–5% of AMLs display biallelic *CEBPA* mutations and display a favorable prognosis; (ix) additional sex comb-like 1 (*ASXL1*) mutations are observed in 10–12% AMLs and are associated with clonal hematopoiesis in elderly persons and with poor prognosis: These mutations represent an early event in leukemogenesis and are associated with chromatin-spliceosome AML group; (x) Wilms tumor protein 1, *WT1*, mutations are observed in 10–13% of AMLs and are associated with *FLT3-ITD* and *CEBPA* mutations and have a poor outcome; (xi) mixed lineage leukemia (*MLL*) mutations are represented by translocations involving the *MLL* gene or partial tandem duplications of the *MLL* gene, are observed in 8–10% of all AMLs and are associated with poor prognosis; (xii) the mutations of serine and arginine splicing factor 2 (*SFRS2*) are observed in 8–10% of all AMLs and make part of the AML class with mutated chromatin; (xiii) the protein tyrosine phosphatase, non-receptor type 11 (*PTPN11*), mutations are observed in 8–10% AMLs and are frequently associated with *TET2, FLT3-ITD* and with normal cytogenetics and (xiv) tumor protein p53 (*TP53*) mutations are observed in 8–14% of all AMLs and are associated with complex karyotype and adverse prognosis [8].

According to the definition of the major molecular events characterizing AMLs, a molecular classification of AML was proposed: (i) AMLs characterized by peculiar molecular events, such as t(15;17) with the fusion gene *PML-RARA* (promyelocytic leukemia/retinoic acid receptor alpha), t(8;21) with the fusion gene *RUNX1-RUNXT1*, inv(16) with the fusion gene *CBFB-MYH11* (core-binding factor subunit beta/myosin 11) and inv(3) with the fusion gene *DEK-NUP214* (DEK-Nucleoporin 214); (ii) the AML chromatin-spliceosome group (18% of total), characterized by mutations of the genes regulating RNA splicing (*SRSF2, SF3B1, U2AF1* and *ZRSR2*), chromatin (*ASXL1, STAG2, BCOR, MLL^PTD^, EZH2* and *FHF6*) or transcription (RUNX1); (iii) the AML group (13% of total) characterized by *TP53* mutations, complex karyotype alterations, cytogenetically detectable copy number alterations or a combination; (iv) NPM1-mutated AMLs, representing 25–30% of all AMLs, with the majority of cases displaying mutations in DNA methylation genes (*DNMT3A, IDH1,* IDH2^R140^ and *TET2*); (v) the *CEBPA* double-mutated AML group, representing about 4% of all AMLs, showing frequent *GATA2* and *NRAS* mutations; (vi) AMLs with *IDH2^172^* mutations, representing 1% of all AMLs, mutually exclusive with *NPM1* mutations; (vii) AMLs with no class-defining genetic alteration, but with at least one driver mutation (about 11% of all AMLs) and (viii) AMLs with apparently no driver mutations (about 4% of all AMLs) [9,10].

The studies of genome sequencing of AML based on the analysis of bulk leukemic cells have provided also some information about the clonal diversity and clonal evolution, as based on the analysis of the frequencies of mutant alleles. Mutations with similar allele frequencies in the bulk samples are assumed to occur simultaneously in the same clone of cells, while mutations that occur at lower frequency are assumed to have occurred later in the evolution of leukemic disease. The comparison of diagnosis and relapse samples allowed the “tracking” of clones from diagnosis to relapse to provide a tentative reconstruction of clonal evolution of AML in the time. Thus, a key study by Welch and coworkers allowed to propose that HSCs/HPCs accumulate benign background mutations as function of age; the initiating mutations occur at a given time in this mutational background and are different for various AML subtype (i.e., *PML-RAR**α* fusion gene for APL, *NPM1* or *DNMT3A* or *IDH1* or *TET2* mutations for M1 AMLs): Since this initial mutational event provides a clonal advantage, all the pre-existing mutations are captured in this initial founding clone [11]. The comparative analysis of diagnosis and relapse samples provided evidence about two major patterns of clonal evolution during AML relapse: (i) The founding clone gained additional mutations and evolved into the relapse clone; and (ii) a subclone of the founding clone survived initial induction/consolidation therapy, gained additional mutations and expanded at relapse, being resistant to therapy [12]. In all relapsing cases, chemotherapy was unable to eradicate the founding clone and induced the accumulation of transversions, probably due to the induction of DNA damage [12].

The combined genetic and functional analysis of purified leukemic cell progenitors from paired diagnosis and relapse samples provided evidence about the existence of two different patterns of relapse: (i) The relapse is dependent on leukemic stem cells with an early differentiation phenotype; and (ii) the relapse is mediated by committed leukemic blasts, that in spite of their mature phenotype, retained stemness features [13]. The link between chemoresistance and leukemic stemness is supported also by other observations, showing that: (i) CD34^+^/CD38^−^ cell frequency predicts the outcome in adult AML [14]; (ii) the stem cell-associated signature allows risk stratification in pediatric AMLs [15] and (iii) the ERG+ enhancer region 85^high^ stem cell marker is associated with resistance to chemo- and radiotherapy predicts AML outcome [16].

The study of AML subtypes allows us to define the mutations that can impact disease outcome and to define the clonal evolution at the molecular level. Thus, the analysis of 10 NPM1-mutated AML patients with a single-cell genetic technique provided evidence that: Clonal architectures were relatively simple in these AMLs, with a number of subclones comprised between one to six; *NPM1* mutations were secondary or subclonal to other mutations, such as *DNMT3A, TET2, WT1* and *IDH2*; in 30% of cases, single-cell analysis of CD34^+^/CD33^−^ cells showed the existence of a leukemic subclone, with one or more driver mutations, but not bearing NPM1 mutations; more than one leukemic subclone engrafted in immunodeficient mice, but in the large majority of cases, the most regenerating subclones displayed *NPM1* mutations; this regenerating subclone was either dominant or minoritarian in the diagnostic sample from which was derived [17].

Another example is given by the study of AMLs characterized by the t(8;21) translocation determining the RUNX1-RUNX1T1 fusion protein. Although this AML subtype is associated with a favorable prognosis, is characterized by a consistent biological heterogeneity, leading to disease relapse in about 40% of patients [18]. At the molecular level, these AMLs were characterized by recurrent mutations in RAS/RTK signaling (64%), epigenetic regulators (45%), cohesion complex (13.6%), MYC signaling (10%) and the spliceosome (8%); on the basis of their high mutant levels and stability at relapse, epigenetic mutations were likely to occur before signaling mutations [18]. Patient age, white blood cell counts, *JAK2, FLT3-ITD*^high^ and *c-KIT*^high^ mutations were associated with a reduced overall survival [18]. The paired analysis of leukemic samples at diagnosis and at relapse allowed to define the clonal evolution of these AMLs: In virtually all patients, part of the individual mutational spectrum was lost and replaced with 1 or >1 new mutations and was allowed to define two groups of relapsing AMLs, one group (group A) with >40% of stable mutations and another one (group B) with >60% of lost/gained mutations [18]. The group B had significantly longer overall survival compared to group A patients [18]. Based on the resemblance of diagnosis and relapse pairs, genetically stable (about 65%) and unstable (about 35%) subgroups can be identified.

Interestingly, Itzykson et al. reported the clonal interference in RAS/RTK pathways, defined as the coexistence of clones sharing a common ancestor clone and harboring independent lesions targeting the same pathway, is observed in about 36% of t(8;21) AMLs and is associated with shorter event-free survival [19]. Greif and coworkers explored the evolution of 50 cytogenetically normal AML patients during therapy and at relapse and obtained evidence of clonal evolution at relapse in 97% of these patients [20]. Of all patients 48% exhibited a gain of new mutations at relapse; 46% of patients with loss or loss + gain of mutations was observed at relapse [20]. Patients with loss of mutations implied the selective disappearance of a sensitive clone and the persistence and outgrowth of a resistant clone and relapsed earlier than patients exhibiting mutation gain [20]. It is of interest to note that the mutations in genes encoding epigenetic regulators are frequently gained at relapse and some of them are associated with resistance toward cytarabine treatment [20]. Finally, this study confirmed that mutations related to clonal hematopoiesis, such as *DNMT3A* mutations, were not cleared during remission [20].

Cocciardi et al. have explored 129 NPM1^mut^ AMLs, at diagnosis and relapse for changes in the most recurrently mutated genes, including *DNMT3A, FLT3, IDH1, IDH2* and *NRAS* and observed that at relapse, a shift in the mutation pattern was observed in 59% of patients; *NPM1* was lost in 9% of patients and *DNMT3A* was the most stable mutation persisting in 95% of cases; *FLT3-ITD* mutation remained stable in 40% of cases, whereas in 35% of cases the *FLT3-ITD* clone changed at relapse and 25% of patients *FLT3-ITD*-positive at diagnosis lost *FLT3-ITD* mutation [21]. These observations suggest that in *NMP1*-mutated AMLs, relapse often originates from persistent leukemic clones; however, *NPM1*^mut^ loss occurring in a minority of cases suggests a second “de novo” or treatment-associated AML [21]. These findings were in line with a recent study carried out in sequentially inducible mouse models showing that *NPM1* mutation causes leukemic transformation and evolution of *DNMT3A*-mutant clonal hematopoiesis; interestingly, the rate of disease progression is accelerated with longer latency of clonal hematopoiesis [22].

Other studies have explored the clonal evolution of FLT3-mutated AML, following relapse after treatment with a third-generation FLT3 inhibitor, such as Giltertinib [23]. Thus, MacMahon and coworkers recently showed through a single-cell DNA sequencing that different patterns of clonal selection and evolution occur in response to FLT3 inhibition, including the emergence of RAS mutations in *FLT3*-mutated subclones, the expression of alternative WT*-FLT3* subclones (or both patterns simultaneously), secondary FLT3-F619L gatekeeper mutations or *BCR-ABL1* fusions [23]. These observations strongly support the need for the development of combinatorial targeted therapeutic approaches for advanced AML [23].

Post-remission therapy in patients with AML is performed either by continuing chemotherapy or by transplantation using either autologous or allogeneic stem cells.

Patients with favorable AML subtypes usually receive chemotherapeutic consolidation or, alternatively, autologous hematopoietic stem cell transplantation (HSCT) in patients without minimal residual disease after induction chemotherapy [24]. Allogeneic HSCT is considered the preferred type of post-remission therapy in poor-, very-poor-risk and intermediate-risk AMLs [21]. In <60 years AML patients, the risk of relapse after allo-HSCT for consolidation after first complete remission is 15–20% for patients with favorable-risk AMLs, 20–25% for intermediate-risk AMLs, 30–40% for poor-risk AMLs and 40–50% for very-poor risk AMLs [24]. Various studies have explored the mechanisms responsible for relapse after allogeneic stem cell transplantation in AML. The 2-year survival rates of patients with AML relapsing after allo-HSCT are <20% due to the limited success of salvage therapies; therefore, there is an absolute need to improve the current strategies to prevent, detect and treat relapse after stem cell transplantation [25].

It is important to underline that the therapeutic effect of allogeneic HSCT is related both to non-immunological and immunological mechanisms, mainly represented by an anti-leukemic effect mediated by donor lymphocytes against patient leukemic blasts. Relapse of the original AML is a major cause of death after allogeneic HSCT for AML. Numerous studies carried out in the last years support the view that relapses may be related to different mechanisms involving not only resistance to chemotherapy of leukemic cells, but also immunological escape of leukemic cells to allogeneic lymphocytes [26].

The pairwise comparison of genetic alterations, including chromosomal alterations and the mutational profile, between leukemia samples obtained at diagnosis and post-transplantation relapse showed an increased number of genomic aberrations after transplantation and in about 10% of patients a deletion of a large homozygous region spanning the whole HLA-locus on chromosome 6p in the relapse sample was observed [27]. Quek and coworkers performed a mutational analysis of disease relapse in a group of allografted AML patients. An increased risk of relapse was observed in AML patients with mutations in *WT1, DNMT3A, FLT3-ITD* and *TP53*, whereas *IDH1* mutations were associated with a reduced risk of relapse [28]. The serial analysis of the mutational landscape from diagnosis to post-transplantation relapse in AML patients providing evidence that 4/9 patients relapsed with re-expansion of pre-existing clones and 5/9 patients relapsed with evidence of subclonal selection [28]. A recent study based on the analysis of 12 relapsing AML patients provided evidence about a significant gain of *WT1* alterations (2/12 pre-transplant and 6/12 post-transplant) and in mutational load [29]. Targeted amplicon sequencing analysis showed that only one of the four cases acquiring new *WT1* mutations at relapse seems to be related to the selection of a pre-existing subclone [29].

Recent studies have provided evidence that in post-transplant relapsing AML patients various immunological escape mechanisms, such as abrogation of leukemia recognition due to loss of the HLA gene, immunosuppression by immune-checkpoint ligand expression, production of anti-inflammatory mediators, release of metabolically active enzymes and loss of pro-inflammatory cytokine production, greatly contribute to leukemia relapse (reviewed in [26]). Thus, Christofer et al. showed that AML relapse after allo-HSCT was not associated with acquisition of relapse-specific mutations in immune-related genes, but to the epigenetic deregulation of some immune pathways, including downregulation of HLS class II genes [30]. Taffalori et al. identified a transcriptional signature specific for post-transplantation relapses, highly enriched in immune-related processes; in parallel, it was documented a deregulation of multiple co-stimulatory ligands, such as PD-L1, CD80 and PVRL2, on AML blasts al post-transplantation relapse and frequent loss of *HLA-DR, -DQ* and *-DP* expression on leukemic cells due to downregulation of the HLA class II regulator CIITA [31]. Furthermore, HLA loss was reported in 12% of patients relapsing after allo HSCT and seems to be an event related to stem cell transplantation in that it is observed in only 0.2–1% of non-transplantation AML cohorts; furthermore, HLA-loss was not observed after matched related-donor HSCT [32]. Finally, Noviello et al. provided evidence that in post-transplantation relapsing patients a higher proportion of bone marrow infiltrating T lymphocytes express inhibitory receptors, such as PD-1, CTLA4, TIM-3 and LAG-3 [33]. Interestingly, some of the immune defects of post-transplantation AMLs can be reversed by IFN-γ or by immune checkpoint inhibitors [31,32].

The presence of CHIP allowed to explore the behavior of clonal hematopoiesis present in donor bone marrow following transplantation to leukemic patients showing basically that HSCT from donors with CHIP is safe and results in similar survival to that of normal donors [34]. Analysis of clones pertaining to clonal hematopoiesis showed engraftment in virtually all cases of these clones; donor-cell leukemia was only rarely observed in recipients; overall survival was not affected by donor CHIP status [34].

The analysis of myelodysplastic syndrome patients relapsing after transplantation provided evidence that in most of these patients small subclones pre-existing before allo-HSCT can drive progression after allo-HSCT, through the new acquisition of structural variants [35].

About 2% of all relapses are represented by late relapses, i.e., relapses occurring >5 years after remission. Longitudinal molecular characterization of patients with a late relapse of AML showed that in most cases the founder leukemic clone persisted after chemotherapy and established the basis of relapse years later; in almost all cases relapsing leukemia acquired at least one relapse-specific mutation [36].

Other studies have explored the clonal mutational dynamics of myelodysplastic syndromes evolving to secondary AMLs. A study of Makishima carried out on a large set of patients showed that during progression the number of mutations, their diversity and clone sizes increased, with alterations frequently present in dominant clones. *FLT3, PTPN11, WT1, IDH1, NPM1, IDH2* and *NRAS* mutations tended to be newly acquired and were associated with more rapid progression to sAML. In contrast, *TP53, GATA2, KRAS, RUNX1, STAG2, ASXL1,* ZRS2 and *TET2* mutations are enriched in high-risk MDS compared to low-risk MDS and had a weaker impact, compared to the other mutations, on sAML progression [37]. The study of leukemic stem cell compartment in MDSs showed a significantly higher subclonal complexity compared to the blast cell compartment and contained large number of age-related variants [38]. Single-cell sequencing studies showed a pattern of non-linear, parallel clonal evolution, with distinct subclones contributing to progression to sAML: Some subclones not detectable in MDS blasts became dominant upon sAML progression [38]. These observations support the view that pre-existing MDS stem cells drive disease progression and transformation to sAML.

### 1.2. Interleukin-3 and Interleukin-3 Receptor

Interleukin-3 (IL-3) is a member of the beta common (β_c_) cytokine family, which also includes granulocyte-macrophage colony-stimulating factor (GM-CSF) and IL-5. This group of cytokines signals through the activation of heterodimeric cell surface receptors, composed by a cytokine-specific α chain and the shared β_c_ subunit [39]. These cytokines signal through the β_c_ subunit: The activation of IL-3R is thought to involve the sequential assembly of a receptor complex, with an initial step represented by the binding of IL-3 to IL-3Rα (also known as CD123), followed by a second step represented by the recruitment of β_c_ subunit and assembly of a receptorial complex, bringing JAK2 molecule together to trigger downstream signaling [40,41].

The extracellular region of IL-3Rα comprises three fibronectin-like (FnIII) domains: Two domains bind IL-3, and a third domain, N-terminal domain (NTD) is highly mobile in the presence of IL-3 and plays a key role in preventing spontaneous receptor dimerization [42]. Thus, the NTD domains exert a double role, protecting from inappropriate receptor signaling and dynamically regulating IL-3R binding and function [42].

IL-3 is a cytokine with multiple biologic functions, mainly produced by activated T-lymphocytes, with an important regulatory activity on the generation of hematopoietic and immune cells [43]. IL-3 is the most pleiotropic cytokine of the β_c_ receptor family, stimulating different myeloid cells. It is functionally different from GM-CSF, in that it also stimulates the production and function of hematopoietic stem cells, mast cells and basophils [44]. The β_c_ cytokines play an important role in emergency hematopoiesis and immunity after infection or injury [44], but their function is dispensable for hematopoiesis in the steady state, as shown by gene knockout studies in mice [45].

The activity of IL-3 is not limited only to the hemopoietic system, but extends also to the endothelial lineage, as supported by the observation that this cytokine is able to stimulate the proliferation of endothelial cells [46].

Many studies have explored the expression of CD123 in hematopoietic cells and particularly at the level of the stem/progenitor cell compartment. These studies were reviewed by Testa et al. [47]. CD123 is expressed on most human CD34^+^ hematopoietic progenitors and its expression is progressively lost during erythroid and megakaryocytic differentiation, while maintained during monocytic and granulocytic differentiation [45]. The studies on normal HSCs have provided conflicting evidences based on the exploration of various sources of HSCs (cord blood or bone marrow) and using different antibodies to detect CD123 expression [47]: Thus, some studies have reached the conclusion that CD123 is not expressed on normal HSCs, while other studies have provided evidence that this receptor is expressed in a part of HSCs [47].

### 1.3. CD123 Is Overexpressed in Many Hematological Malignancies

Studies carried out in the last two decades have shown that IL-3Rα is overexpressed in many hematological malignancies. Many studies have explored in detail the pattern of expression of CD123 in AMLs and in blastic plasmacytoid dendritic cell neoplasm (BPDCN). Initial studies on AMLs have led to the observation that CD123 is overexpressed on stem/progenitor leukemic cells CD34^+^/CD38^−^, while normal CD34^+^/CD38^−^ normal stem cells apparently do not express CD123 [48]. A subsequent study based on the screening of CD123 expression in various hemopoietic malignancies showed that this receptor chain is frequently expressed at high levels in AMLs and B-ALLs [49]. Testa et al. explored a large set of AML patients and reported that 45% of these patients overexpress IL-3Rα [50]. Importantly, these authors showed that CD123 overexpression on leukemic blasts was associated with increased cycling activity, increased cellularity at diagnosis, hypersensitivity to IL-3 stimulation (with increased Stat5 activation) and poor prognosis [50].

Wittwer et al. have confirmed that high CD123 levels enhance proliferation of leukemic blasts in response to IL-3 but showed also that CD123 overexpression induces a downregulation of CXCR4 [49]. CXCR4 is the receptor of stromal-derived growth factor-1 (SDF-1) and plays an essential role in the regulation of HSC homing and migration. Thus, it was hypothesized that the CD123 overexpression, through CXCR4 downregulation, may induce the egress of BM AML leukemic stem cells (LSCs) into the circulation [51].

Arai and coworkers have explored CD123 expression on 48 de novo AMLs by immunohistochemistry and reported that CD123 expression on AML blasts was associated with a failure to achieve a complete response to initial induction chemotherapy and poor overall survival [52].

Other studies have attempted to define the immunophenotypic and molecular properties of AMLs overexpressing CD123. Thus, Testa and coworkers provided evidence that AMLs overexpressing CD123 display some peculiar immunophenotypic features (low CD34 expression and high CD11b and CD14 expression) and are frequently associated with FLT3-ITD mutations [53,54]. Interestingly, a subset of AMLs overexpressing FLT3, in the absence of mutations of this receptor, overexpress also CD123 and may represent a peculiar AML subset, whose proliferation was driven by the overexpressed CD123 and FLT3 [55].

Rollins-Raval and coworkers have confirmed these findings and have reported CD123 overexpression (by immunohistochemistry) in 83% of FLT3-ITD-mutaed AMLs and in 62% of AML cases with mutated NPM1 [56]. Brass and coworkers have confirmed these findings through a detailed flow cytometric analysis of CD123 in a very large set of AMLs, showing that CD123 was expressed in the large majority of AMLs, with low expression in erythroid and megakaryocytic leukemia, higher CD123 expression in FLT3-ITD-mutated and NPM1-mutated AMLs [57]. Importantly, two studies reported IL3Rα expression at the level of the CD34^+^CD38^−^ cell fraction in FLT3-ITD-mutated AMLs [58,59]. Angelini et al. provided a detailed immunophenotypic characterization of these cells, displaying also CD25 and CD99 expression [59].

Ehninger et al. have explored CD123 and CD33 expression in a cohort of 319 AML patients and showed that AMLs with adverse cytogenetics express CD123 at levels comparable to those with favorable and intermediate subtypes [60]. They confirmed also the high expression of CD123 on FLT3-ITD and NPM1-mutated AMLs [60].

CD123 was shown to be overexpressed in a rare group of undifferentiated AMLs, resembling AMLs with minimal differentiation [61].

Many studies have explored CD123 expression at the level of leukemic progenitor/stem cells, mainly contained in the CD34^+^/CD38^−^ cell fraction. Thus, Guzman and coworkers provided evidence that CD34^+^/CD38^−^/CD123^+^ cells are able to initiate a leukemic process when inoculated into immunodeficient mice [62]. CD34^+^/CD38^−^/CD123^+^ cells were clearly detectable in about 75% of AML patients [63] and their number is predictive of the clinical outcome [64]. More recent studies have explored CD123 expression on the leukemic stem population in the context of the expression on these cells of other membrane markers. Several membrane markers, including CD123, CD33, CLL1, TIM3, CD244, CD47, CD96, CD157 and CD7 were ubiquitously expressed on AML bulk cells at diagnosis and relapse, irrespective of genetic features [65]. Haubner et al. have explored the expression of these membrane antigens on a large set of AMLs at diagnosis and at relapse. CD33, CD123, CLL1, TIM3, CD244 and TIM3 were ubiquitously expressed on AML bulk cells at diagnosis and relapse, irrespective of genetic features [66]. Importantly, for the clinical implications, CD33 and CD123 are homogenously expressed at relapse; CD123 was more expressed than CD33 at the level of LSCs [66].

In another study, Yan and coworkers provided evidence that CD123, as well as CD47 expression on leukemic blasts is related not only to stemness, but also to chemoresistance [65]. The histone deacetylase inhibitor romidepsin reversed the gene expression profile of CD123^+^ chemoresistant leukemic cells and efficiently targeted chemoresistant leukemic blasts in xenograft mouse models [67]. In line with this observation, the presence of CD34^+^/CD38^−^/CD123^+^ cells de novo AMLs negatively impacts in disease free survival (DFS) and overall survival (OS) [68].

Recent studies support the evaluation of CD123 expression as a marker of minimal residual disease (MRD). MRD can be defined as the persistence after therapy of a small burden of leukemic cells, not detectable by standard histopathological criteria (bone marrow morphology); the presence of MRD after therapy is a negative prognostic marker of increased risk of relapse and shorter survival in AML patients [69]. Its detection is of fundamental importance to risk stratification and to define additional treatments [69]. MRD can be assessed through the detection of leukemia-specific molecular abnormalities or though flow cytometry of bone marrow cells [69]. Thus, quantification of NPM1-mutated transcripts after therapy in patients with NPM1-positive AMLs provided a valuable tool to assess MRD and a powerful prognostic information independent of other risk factors [70]. MRD can be detected also by flow cytometric analysis of bone marrow cells using a panel of antigens aberrantly expressed in AML blasts, including CD123 [71]. The evaluation of both molecular and immunophenotypic markers probably represents the best methodology to assess MRD and its evaluation is of paramount importance in the choice of the treatment strategy for each AML patient [72].

These studies imply some considerations about the relationship between clonal evolution and CD123 expression in AMLs. As above discussed, during therapy, the AML cell populations may evolve by either acquiring additional mutations responsible for drug resistance or by losing mutations associated with sensitivity to the treatment (following a process of linear evolution) or by outgrowth of a secondary clone (subclone) following eradication of the major clone at diagnosis (following a process of branching evolution). Therefore, AML cell populations at relapse may have evolved from either clonal or subclonal cell populations, present at diagnosis, with potential acquisition of additional mutations. Thus, either through linear or branching evolution, leukemic cells undergo a process of adaptive clonal evolution to survive to new environmental conditions. These processes of dynamic clonal evolution imply a consistent clonal heterogeneity of AMLs and the extreme difficulty to eradicate the leukemic process through the targeting of a single genetic abnormality. At variance with molecular markers involved in the dynamics of clonal evolution, CD123 is equally expressed in AML bulk cells and leukemic stem cells at initial diagnosis and relapse and therefore its expression is not related to leukemic clonal evolution [66]. The independency of CD123 expression from AML clonal evolution strongly supports CD123 as a potential therapeutic target of AMLs at various disease stages: Initial diagnosis, MRD and relapse. 

Blastic plasmacytoid dendritic cell neoplasm (BPDCN) represents the malignancy with the most promising clinical applications of CD123 targeted therapy. BPDCN is an extremely rare clonal hematological malignancy of plasmacytoid dendritic cell precursors, usually affecting elderly males and presenting in the skin with frequent involvement of the bone marrow, peripheral blood and lymph nodes [73,74]. The epidemiology, pathology, molecular abnormalities and clinical features of BPDCN were recently reviewed [73,74].

Two historical studies have considerably improved the understanding of the physiopathology of this rare neoplasia and have provided the key data for the identification of CD123 as a potential therapeutic target. Thus, Lucio and coworkers have for the first time shown a constant high CD123 expression and have proposed plasmacytoid dendritic cells as the cells of origin of BPDCN [75]. These findings were confirmed by Chaperot et al., showing immune function in leukemic cells similar to plasmacytoid dendritic cells [76]. These authors showed also that CD4^+^/CD56^+^ cells express elevated levels of the IL-3Rα chain: Following incubation with IL-3, the leukemic cells undergo a partial maturation and became a strong inducer of naïve CD4^+^ T cell proliferation [76].

The patients with BPDCN have a negative prognosis when treated with standard chemotherapy. The only treatment inducing durable remissions is the high-dose chemotherapy, followed by allogeneic stem cell transplantation [77]. In a large series of 43 patients, the median overall survival was of 8–9 months following treatment with chemotherapy and about 23 months following chemotherapy and allogeneic stem cell transplantation [78].

Other studies have shown the overexpression of CD123 in some lymphoid malignancies. The screening of B-cell malignancies provided evidence that in patients with lymphoproliferative disorders of mature B-lymphocytes was positive only in hairy cell leukemia [49]. Subsequent studies have confirmed this initial observation, showing also the analysis of CD123 expression helps also to distinguish between hairy cell leukemia and hairy cell leukemia-variant [79,80,81]. The characteristic immunophenotype CD19^+^, CD20^+^, CD11c^+^, CD25^+^, CD103^+^ and CD123^+^ is diagnostic for hairy cell leukemia [82].

While virtually all lymphomas were negative for CD123 expression, neoplastic cells of Hodgkin lymphoma were frequently CD123^+^ [83].

Other studies have explored in detail CD123 expression in acute lymphoid leukemia (ALLs). Initial studies have shown that B-ALL, but not T-ALL frequently overexpress CD123 [49,50]. This finding was confirmed in subsequent studies, showing also that CD123 overexpression in B-ALL associates with hyperdiploid genotype [84]; furthermore, CD123 is clearly expressed at higher levels in B-ALL blasts than in normal B-lymphoid progenitors/precursors [85,86]. More recently, Angelova et al. have performed a large screening of CD123 expression on a large set of ALLs, showing that CD123 expression was more prevalent in Philadelphia chromosome-positive patients than in Philadelphia chromosome-negative patients [87]. Importantly, Liu et al. reported the existence of a negative correlation between CD123 expression on B-ALL blasts and both disease-free and overall survival [88].

## 2. Therapeutic CD123 Targeting

### 2.1. SL-401 (Tagraxofusp)

Initial studies of CD123 targeting were based on the use of the natural ligand IL-3, fused with a cytotoxic drug. To this end, a genetically engineered fusion toxin was generated, composed of the first 388 amino acid residues of diphtheria toxin (DT) with a His-Met (H-M) linker, fused to human IL-3. This DT_388_ exerted a marked cytotoxic effect on CD123^+^ leukemic blasts [89] and was tolerated in primates in vivo up to 100 μg/Kg [90,91]. The level of cytotoxicity exerted by DT_388_ on leukemic blasts was directly related to the level of IL-3Rα/IL-3Rβ expressed on leukemic cells [92]. Importantly, DT_388_ was found to be cytotoxic for cells with properties of leukemic stem cells [93]. The fusion of DT to a variant of IL-3 displaying increased binding affinity (IL-3[K116W]) resulted in the generation of a fusion protein DT_388_IL-3[K116W] more active than DT_388_IL-3 in inducing the killing of leukemic blasts [94].

This fusion protein named SL-401 (tagraxofusp) was introduced in phase I/II clinical studies for the treatment of BPDCN, AML and other hematological malignancies. Preclinical studies have strongly supported the clinical use of SL-401 in BPDCN. BPDCN is an aggressive hematologic malignancy derived from the malignant transformation of plasmacytoid dendritic cells. BPDCN represents a distinct disease entity in the group of AMLs and related precursor neoplasms. This neoplasia is characterized by high ubiquitous expression of CD123 and represents an ideal candidate for therapeutic targeting with SL-401. In this context, preclinical studies showed that SL-401 exerted a marked cytotoxic effect against primary BPDCN blasts, that are more sensitive to this drug than AML or ALL primary blasts; SL-401 was more toxic against BPCDN cells than other chemotherapeutic agents; treatment with SL-401 increased the survival of immunodeficient mice inoculated with BPDCN cells [95].

The first clinical trial of SL-401 in BPDCN involved the treatment of 11 patients in the context of a phase I study [86]. The maximum tolerated dose was 12.5 μg/Kg/day; the drug administration was relatively well tolerated, and the reported adverse events were grade 3/4; 78% of patients displayed an objective response, with 55% complete responses after the first cycle of treatment [96]. In 2017, the results of a phase II clinical study involving the treatment of 32 BPDCN patients were presented, showing 84% of objective responses, with 59% of complete responses [97]. A part of responding patients was bridged to stem cell transplantation after a durable response from SL-401 treatment [97].

In December 2018, the. Food and Drug Administration (FDA) approved SL-401 for adult and pediatric BPDCN on the basis of the results of a recently published open-label, multicohort phase III study in which 47 patients with untreated or relapsed BPDCN received an intravenous infusion of 7 or 12 μg/Kg body on days 1 to 5 of each 21-day cycle. Of the 47 patients, 29 received 12 μg of tagraxofusp as the first-line treatment and 15 as the second- and third-line of treatment [98]. Among the 29 untreated patients, 72% achieved a complete response, 45% went on to undergo stem cell transplantation, with a survival rate of 52% at 24 months; among the 15 previously treated patients, the response rate was 67% and the median overall survival was 8.5 months [98]. Capillary leak syndrome was observed in 19% of the patients and was associated with two deaths [98]. The results of this study provided the basis for a targeted therapy option for patients with BPDCN. 

SL-401 was also tested in few pediatric BPDCN patients, providing preliminary evidence of clinical activity [99].

In addition to the above-mentioned studies, additional more recent observations support the rationale of using SL-401 as an IL-3Rα targeting agent in AML patients. Thus, tagraxofusp induced potent cytotoxic activity against CD123-positive AMLs and myelodysplastic syndrome blast cells; furthermore, it exerted also some cytotoxic activity against normal hemopoietic progenitor cells [100]. Thus, this drug was active against hemopoietic cells expressing high or low CD123 levels. Tagraxofusp seemed also able to overcome resistance mechanisms, as those related to the “protection” exerted by stromal cells, reducing CD123 expression on leukemic blasts [98]. Tagraxofusp administration improved the survival of immunodeficient animals xenografted with primary AML cells [100]. For these properties, tagraxofusp was considered as a potential mean to create a bridge to stem cell transplantation [100].

Thus, as above discussed, AMLs represent another potential therapeutic target of SL-401. The results of a phase I clinical trial performed on 45 AML patients, receiving one single infusion of SL-401, showed that the maximum tolerated dose was 12.5 μg/Kg/day, no toxicity grade IV/V adverse events were observed, grade II/III adverse events included fever, hypoalbuminemia, transaminitis, hypotension and hypocalcemia, and few responding patients were observed (one complete response and two partial responses) [101]. Another approach for the therapeutic use of SL-401 in AML consists to treat the minimal residual disease (MRD), usually treated with the traditional consolidation therapy. Thus, a phase I/II clinical trial (NCT02270463) is ongoing in patients with AML who are at risk of relapse and are not candidates for allogeneic stem cell transplantation [102]. The preliminary results of this trial presented at the 2017 American. Society of Hematology (ASH) Meeting support the safety of the treatment, but the results about the possible efficacy were not yet available and will require additional time to be evaluated [102].

Stephansky and coworkers have explored the possible mechanisms of resistance of leukemic cells to CD123 targeting by SL-401 and observed that a mechanism not dependent upon CD123 expression was responsible for resistance. In fact, these authors showed that resistant leukemic cells downregulated the expression of DPH1, the enzyme that converts histidine 715 on eEF2 to diphthamide, the direct target of ADP ribosylation by diphtheria toxin [103,104]. DPH1 expression was decreased in leukemic cells of patients treated with SL-401 and resistant to this drug; furthermore, DHP1 downmodulation in leukemic cells decreased their sensitivity to SL-401. Interestingly, the DNA methyltransferase inhibitor azacytidine reversed this effect and acted in cooperation with SL-401 to promote the death of AML blasts [103,104]. These observations have supported the development of a clinical trial (NCT 03113643) involving the administration of SL-401 in association with azacytidine at standard dose to refractory/relapsed AML or high-risk myelodysplasia patients.

Few studies have supported the use of SL-401 in the therapy of multiple myeloma. These studies were based on the strategy of targeting the bone marrow microenvironment that in multiple myeloma promotes disease development and progression. Plasmacytoid dendritic cells (PDCs) are increased in bone marrow of multiple myeloma and highly express IL-3R. Targeting of PDCs in bone marrow milieu with SL-401 decreases viability of PDCs, blocks PDC-induced multiple myeloma growth and synergistically increases anti-tumor activity of bortezomib and pomalidomide [105]. Furthermore, IL-3R promotes the progression of osteolytic bone disease in multiple myeloma [105]. Finally, and importantly, SL-401 decreases the viability of cancer stem-like cells in multiple myeloma [105]. These observations support the experimental use of SL-401 in the treatment of multiple myeloma. SL-401 is currently under evaluation in combination with pomalidomide and dexamethasone in relapsed/refractory multiple myeloma patients [106]. The first results of this ongoing clinical trial showed that SL-401 is relatively well tolerated in this therapeutic setting and shows some signs of clinical activity [106].

Other clinical studies are exploring the possible clinical activity of SL-401 in myeloproliferative disorders. A preclinical study showed IL-3R expression in systemic mastocytosis, clonal eosinophilia and a minority population of myelofibrosis [107]. SL-401 is under evaluation in patients with advanced high-risk myeloproliferative disorders, including systemic mastocytosis, myelofibrosis, primary eosinophilic disorders and chronic myelomonocytic leukemia (CMML) [105]. Preliminary results in this clinical study showed that SL-401 administration is relatively well-tolerated and induced the secondary/adverse events observed in other clinical studies with this drug [108]. On 2018, the results on 20 CMML patients were presented, showing a marked decrease of splenomegaly. Bone marrow complete responses were observed in 30% of patients [109]. Some patients were bridged to stem cell transplantation [109].

Results from 23 patients with myelofibrosis were recently reported, showing that tagraxofusp treatment elicited a spleen response in 56% of patients with a spleen size ≥5 cm at baseline; interestingly, 100% of patients with monocytosis showed a spleen response [110]. Furthermore, treatment with tagraxofusp was associated with an improved quality of life.

Interestingly, a recent study reported the existence of clonal plasmacytoid CD123^+^ dendritic cells in about 20% of CMML patients [108]. Exome sequencing studies showed that these cells have the typical Ras mutations observed in CMML [111]. An excess of plasmacytoid dendritic cells correlates with regulatory T cell accumulation and an increased risk of acute leukemia transformation [111].

As above reported, the K116W variant DT_388_-IL-3 molecule (DT_388_-IL-3[K116W] or SL-501) displayed greater IL-3R binding affinity and cytotoxic activity against AML blasts than SL-401 and possesses low activity against normal hematopoietic progenitor/stem cells [82,83]. A preclinical study showed that SL-401 deplete chronic myeloid leukemia (CML) stem cells and may be used as a strategy to improve the effectiveness of current CML therapy, mainly targeting tumor bulk using tyrosine kinase inhibitors [112].

The Stemline Therapeutics Inc. developed also SL-101, an anti-IL-3R antibody-conjugate in which the single-chain fragment variable (scFv) regions of the antibody were genetically fused to a truncated *Pseudomonas* exotoxin containing its translocation and ADP-ribosylation domains [113]. 

Han and coworkers have explored the anti-leukemic activity of this antibody-conjugate against AML blasts. In a retrospective analysis of 86 newly diagnosed AML patients, these authors showed that a higher proportion of CD34^+^CD38^−^CD123^+^ leukemic stem cells at remission stages was associated with persistent MRD and predicted shorter progression-free survival (PFS) in patients with poor-risk cytogenetics [113]. SL-101 was shown to suppress the function of leukemic progenitors, while sparing normal hemopoietic progenitor cells; in xenograft AML models the repopulating capacity of LSCs pretreated with SL-101 in vitro was significantly inhibited [113].

### 2.2. CSL362 (Taclotuzumab)

The optimal properties of an IL-3R mAb would consist in inhibiting the binding of IL-3 and in activating innate immunity. An antibody responding these two properties was the anti-CD123 mAb 7G3 capable of inhibiting IL-3-mediated proliferation of leukemic cells [114] and of impairing leukemic stem cells in vivo [115]. The 7G3 mAb was humanized and affinity-matured and was engineered in its Fc-domain to improve its cytotoxicity against AML cells: the antibody thus modified was called CSL362 and displayed the same capacity to neutralize IL-3, but displayed increased antibody-dependent cytotoxicity against AML cells compared to 7G3 [116].

For its properties, the CSL 362 antibody entered a plan of clinical development. Pharmacodynamic studies provided evidence that CSL 362 administration to immunodeficient mice xenografted with human AML cells significantly enhanced the antileukemic effect induced by cytarabine/daunoribicin [117]. The administration of CSL 362 induced a dose-dependent decrease of peripheral basophils and plasmacellular dendritic cells, but no significant effects on hemopoietic progenitor/stem cells were observed [118].

A preclinical study in CML showed that CSL 362 was able to markedly reduce the engraftment of CML cells, due to the killing of CD123^+^ LSCs [109]. In mice treated with CSL 362 antibody-dependent cell-mediated cytotoxicity (ADCC)-facilitated lysis was mediated mainly by few CML autologous natural killer (NK) cells [119]. Additional preclinical studies definitely showed that CSL 362 in vitro induces ADCC-dependent lysis of AML blasts, as well as of LSC-enriched CD34^+^/CD38^−^/CD123^+^ cells and in vivo reduces leukemic cell growth in AML xenografts in immunodeficient mice [120]. The study of the three-dimensional structure of CD123 allowed us to define the molecular mechanism through which CSL-362 inhibits IL-3R. CSL362 binds to the N-terminal domain (NTD) of IL-3Rα: This domain allows the optimal IL-3 binding to its receptor and is present in two distinct conformations, open and closed [121]. CSL362 can utilize functionally distinct binding sites on IL-3R and can inhibit IL-3 signaling without inhibiting IL-3 binding [121].

Lee et al. explored in more detail the effect of CSL362 on AML cells in suitable preclinical models and provided clear evidence that CSL362 prolonged the survival of immunodeficient mice xenografted with AML cells only when administered together with chemotherapy (cytarabine/daunorubicin), but not when administered alone [122].

CSL362 was tested in a phase I clinical study (NCT01632852) in a group of 40 refractory/relapsing AML patients, showing a clinical response in only two patients, suggesting that the use of this drug alone is not sufficient for the treatment of refractory AML [123]. A second phase I study (NCT01272145) was carried out in a group of AML patients who have achieved a first or a second remission, but who are not candidate for stem cell transplantation and have a high risk of disease relapse [124]: 11 of these patients displayed MRD at baseline and four of these 11 patients converted to negativity after treatment with CSL362; at week 24 after treatment, these four patients remained in complete remission, while the remaining seven patients relapsed before week 24 [125]. The maintenance of an MRD^−^ condition may suggest eradication of the leukemic disease [124]. In more recent clinical trials, the CSL362 antibody was renamed talacotuzumab. Platzbecher et al. recently reported the results of a clinical study based on talacotuzumab administration to elderly, high-risk AML or MDS patients who have failed treatment with hypomethylating agents [126]. 24 AML patients, with a median age of 77 years were explored for response to talacotuzumab and five of these patients displayed a hematological improvement, corresponding to a partial response [127]. These observations suggest a limited benefit of this drug in this patient setting, a phenomenon related also to the compromised immune profile present in these patients (reduced mature NK cells, increased expression of inhibitory NK-cell receptors) [128].

A clinical study is evaluating the efficacy and safety of decitabine plus talacotuzumab versus decitabine alone in AML patients ineligible for intensive chemotherapy.

A factor limiting the efficacy of CSL362 in vivo is the limited number of NK lymphocytes present in some cancer patients. To bypass this limitation, Ernst and coworkers have explored the effectiveness of combining CSL362 with human allogeneic NK cells to kill Hodgkin lymphoma cells showing that this combination was highly effective in killing CD123^+^ lymphoma cells and that CSL362 facilitated NK cell antibody-dependent cell-mediated cytotoxicity (ADCC) of Hodgkin lymphoma targets in ARF6/PLD-1 [129].

### 2.3. IMGN632: A CD123-Targeting Antibody Drug-Conjugate

Recently, the development of a CD123-targeting antibody-drug conjugate (ADC) was reported, which is composed by a humanized anti-CD123 antibody G4723A linked to a DNA mono-alkylating payload of the indolinobenzodiazepine pseudodimer (IGN) class of cytotoxic compounds (called IMGN632) [117]. The activity of IMGN632 was compared to that of X-ADC, the ADC involving the G4723A antibody linked to a DNA crosslinking IGN payload [130]. Both IMG632 and X-ADC exerted both in vitro and in vivo a potent antileukemic effect, but IMGN632 was >40 fold less cytotoxic to the normal myeloid progenitors than X-ADC [130]. Importantly, IMGN632 exerted anti-leukemic effects at doses well below those causing cytotoxic effects to myeloid progenitors [130]. Finally, IMGN632 exerted a potent anti-leukemic effect in various AML xenograft models [130]. These observations strongly support the clinical development of IMGN632 [130].

Another recent preclinical study provided evidence that IMGN632 was active in promoting the killing of B-ALL blasts [87]. Thus, Angelova and coworkers provided evidence that CD123 expression was more prevalent in B-ALL than in T-ALL; furthermore, within B-ALLs, CD123 expression was more pronounced in Philadelphia chromosome-positive patients [87]. IMGN362 resulted to be highly cytotoxic to B-ALL, with a half-maximal inhibitory concentration corresponding to the range 0.6–20 pM [87]. These observations support a possible clinical use of IMGN632 for B-ALL targeting.

In line with this study, it was recently reported that IMGN632 exerted in vivo a pronounced efficacy against patient-derived xenografts (PDXs) derived from a wide range of ALL subtypes, expressing various levels of CD123 [131].

A recent preclinical study explored the targeting of CD123 in BPDCN using IMGN632, showing significant anti-leukemic effects, even at a dose of IMGN362 10-fold lower than the anticipated therapeutically active dose [132]. Another recent preclinical study showed a synergism between poly (ADP-ribose) polymerase (PARP) inhibition (using olaparib or other PARP inhibitors under development) to enhance the therapeutic efficacy of IMGN362 across different primarily refractory/relapsed AML samples [133].

IMGN362 is currently in phase I evaluation for relapsed/refractory CD123-positive hematological malignancies (NCT 03386513). Recently, the initial safety and antileukemia activity findings from the dose-escalation stage of this trial were reported, based on the analysis of 12 patients (five with relapsed/refractory disease and six at first relapse; seven patients had adverse cytogenetics and 50% had secondary AML) [134]. The administration of the drug was tolerated, and no major adverse events were observed [134]. Of the treated patients 33% achieved a complete response and these data support the continuation of this study and further clinical exploration of IMGN362 [134].

Interestingly, a recent study explored a possible anti-leukemic synergism of IMGN362 with the BCL-2 inhibitor venetoclax showing that the combination of the two drugs increases the killing of primary AML blasts; in vitro studies on various AML cell lines indicate additive/synergistic effects of the two drugs and in vivo the combination of the two drugs clearly prolongs survival and increases anti-leukemic activity in various AML PDX models [135]. These observations strongly support the clinical exploration of the combination of these two drugs in a clinical trial in AML patients.

### 2.4. Bispecific CD123 Monoclonal Antibodies

Kuo and coworkers have reported the development of a bifunctional fusion anti-CD123 and anti-CD3 (CD123 × CD3 bispecific scFv) [136]. The fusion antibody exhibited several interesting properties, compared to the monospecific anti-CD123 antibody: (i) An increase of target cell-binding affinity; (ii) increased in vivo stability, as evidenced by enhanced serum half-life; (iii) induction of T-cell-mediated target cell killing [136]. Subsequently, Hussuini et al. developed a bispecific antibody, composed by the V^H^ of one antibody in tandem with the V^L^ of the other antibody, with specificity of interaction with the N-extracellular domain of CD123 and the extracellular domain of CD3 in the human T cell receptor complex [137]. The incubation of this CD3xCD123 dual-affinity re-targeting (DART) with AML leukemic blasts induced both activation of T lymphocytes and killing of leukemic blasts [137]. The infusion of the bispecific antibody into immunodeficient mice xenografted with AML blasts resulted in total clearance of peripheral blood leukemic cells and subtotal elimination of leukemic cells in bone marrow [137].

In 2015, the MacroGenics (Rockville, MD, USA) reported the development of MGD006, a novel 589 kDa CD3 × CD123 DART protein produced in Chinese hamster ovary cells [129]. The CD3 × CD123 DART molecule was composed of humanized mouse anti-human CD3 and anti-human CD123 Fv sequences [138]. MGD006 exhibited a potent anti-leukemic activity both in vitro and in vivo and was well tolerated in monkeys in continuous infusion [138]. Al-Hussaini and coworkers reported an extensive characterization of this DART antibody, showing that: (i) CD3 × CD123 induces effector-target cell interaction and promotes T cell activation and proliferation; (ii) CD3 × CD123 induces antigen-specific cytotoxicity against CD123-expressing cell targets and, particularly, AML blasts, by donor T lymphocytes; (iii) CD3 × CD123 DART induces killing of AML blasts also in the presence of stroma; (iv) in vitro studies showed a minimal effect of CD3 × CD123 DART on CD34^+^ progenitors and a reduction of CD14^+^/CD123^+^ monocytes and (v) CD3 × CD123 exerts a robust anti-leukemic activity in vivo [139]. These observations strongly supported the evaluation of this CD3 × CD123 DART in phase I clinical trials in refractory/relapsing AML patients.

MDG-006, with the commercial name of flotatuzumab is being evaluated in phase I/II clinical studies. A phase I study showed that the recommended phase II dose was 500 ng/kg/day. Thirty AML patients are mostly with primary refractory disease and with high-risk disease [140]. Infusion-related cytokine syndrome occurred in 13% of patients and was managed with standard supportive care [140]. Anti-leukemic activity was detected in 67% of evaluable patients with 19% of complete responses [140]; importantly, the complete response rate was 31% among patients with refractory AML, but only 0% among those with relapsed AML [140]. On the basis of these preliminary observations, enrollment to this study was expanded to better define the anti-leukemic activity of flotetuzumab in refractory AML patients and to try to define biomarkers to predict and identify patients more likely to respond to this drug.

Since a previous study suggested that AML patients with an immune-enriched tumor microenvironment, as supported by the detection of an increased expression of genes associated with CD8 lymphocytes, B cells and Th1 cells, CXCL9 and CXCL10, are less likely to respond to anthracycline-based cytotoxic chemotherapy and have a shorter relapse-free survival [141], it was hypothesized that the presence of an immune-enriched gene signature in the bone marrow of AML patients more likely to respond to flotetuzumab-based immunotherapy [142].

The study of patients treated with flotetuzumab allowed to identify the frequency of CD4^+^ cells at baseline as a potential biomarker for identifying patients with a higher risk of developing more severe cytokine release syndrome [143]. Early use of tociluzumab (anti-IL-6 mAb) blunted the severity of cytokine release syndrome and did not affect the response to treatment [143].

Given the short circulating half-life, flotetuzumab needs to be administered as a continuous infusion, thus determining prolonged exposure times, with limited Cmax changes. The use of more stable, long-acting Fc-bearing, CD3-interacting antibodies would abrogate the need for continuous supply but would expose to the risk of cytokine release syndrome due to the high Cmax levels required to maintain appropriate concentrations over several days [144]. Thus, the development of CD3-engaging DART molecules with reduced affinity for CD3 that maintained maximal target cell killing and T cell proliferation, allowed the production of CD3 × CD123 and CD3 × CD19 DART molecules, acting at higher concentrations than wild-type counterpart, but with reduced cytokine releasing capacity [144].

Very recently, Ravandi and coworkers reported the development of a bispecific monoclonal antibody, XmAb14045 targeting both CD123 and CD3 and stimulating targeted T cell-mediated killing of CD123-expressing cells; this antibody is a full-length immunoglobulin molecule requiring intermittent infusions [145]. Using this molecule, a phase I clinical study was performed and the results on the first 64 treated patients (63 with refractory/relapsed AML and 1 with ALL) were recently reported [145]. Of treated patients 77% experienced a cytokine release syndrome and 11% of grade ≥3. In part A of the study, single agent anti-leukemic activity was evidenced with three complete responses in 23% of treated patients; two responders were bridged to stem cell transplantation and the third remained in remission at week 14+ after initiating therapy [145].

Recently, the development of a dual-targeting triplebody 33-16-123 (SPM-2) agent, with binding sites for target antigens CD33 and CD123, and for CD16 to engage NK as cytolytic effectors was reported [146]. Primary blasts of most AMLs carry at least one of these antigens; blasts from 29 AML patients were lysed at nanomolar concentrations of SPM-2, the optimal lytic effect being observed for leukemic cells with a combined density of CD33 and CD123 above 10,000 molecules/cell [146]. Cell populations phenotypically enriched in leukemic stem cells (CD34^+^CD38^−^) carry increased CD33 and CD123 expression and were lysed even at low SPM-2 concentrations [146].

### 2.5. Chimeric Antigen Receptor (CAR) T Cell Therapy Targeting CD123

Recent studies of anti-tumor adoptive cellular therapies have explored the potential therapeutic impact of genetically engineered cells. In this context, two types of genetically engineered T cells have been developed: (a) T cell receptor-engineered T lymphocytes that recognize a specific antigen in the context of human leukocyte antigen (HLA) receptors (HLA restricted) and (b) chimeric antigen receptor (CAR) transduced T cells that recognize membrane antigens in an HLA-independent and antibody-specific manner. CAR T cells display a combination of various important biologic properties related to the presence of an antibody in terms of the specificity and affinity in the recognition of a target antigen and of T lymphocytes endowed with a long-term biologic function and with a consistent capacity of in vivo circulation [147,148]. CARs are artificial receptors composed of three domains: (i) An extracellular antigen-binding specific domain derived from an antibody’s single chain variable fragment; (ii) a hinge and transmembrane fragment and (iii) an intracellular T-cell signaling domain [147,148]. Furthermore, CAR T cells can be distinguished into first-, second- and third-generation CAR T cells according to the presence of co-stimulatory signals near to the zeta-signal-transducing subunit of the TCR/CD3 receptor complex, and consisting in one or two additional co-stimulatory molecules, such as CD28, CD27, DAP-12 and CD137 [147,148]. These changes contribute to modify the proliferation, survival in vivo and activation of CAR T cells [147,148].

The approval of the anti-CD19 CAR T cell product tisagenlecleucel by FDA and European Medicinal Agency (EMA) for the treatment of relapsed pediatric B-ALL represents the start of a new era in acute leukemia therapy. This decision was based on the data of a phase II trial reporting the results on 75 B-ALL patients treated with tisagenlecleucel, with complete remissions in 81% of patients at three months, and event-free survival rates of 73% and 50% at 6 and 12 months, respectively [149]. Unfortunately, the results obtained in B-ALL have not yet been translated at the level of AML, where the progress was hampered by the difficulty to find a suitable targetable membrane antigen [150]. Recent preclinical and clinical studies are exploring the possible role of CD123 as a target of specifically engineered CAR T cells.

In 2013, Tettamanti and coworkers have transduced cytokine-induced killer (CIK) cells with a retroviral vector (a first generation CD123 CAR) encoding an anti-CD123 CAR; transduced CIK cells were able to kill AML cell lines and primary AML leukemic blasts, including leukemic progenitor/stem cell populations [151]. The same group reported in 2014 a third generation CD123 CAR containing CD248 and OX490 as co-stimulatory molecules: CIK cells transduced with this vector exhibited potent anti-leukemic activity and had only a limited inhibitory effect on normal bone marrow progenitor/stem cells in xenograft models [152].

Mardiros and coworkers reported the development of CARs containing a CD123 specific scFv in combination with a CD28 co-stimulatory domain: T lymphocytes obtained from AML patients were modified to express CD123 CARs and acquired the capacity to lyse autologous AML blasts in vivo in a xenogeneic model of AML [153]. These CD123 CARs did not kill normal hemopoietic progenitors.

Gill and coworkers using a second generation CD123 CAR, containing 4-1BB co-stimulatory domain, showed efficient elimination of leukemic cells in a xenograft model of AML [154]. However, these authors reported a marked inhibition of normal hematopoiesis in a model of hematopoietic reconstitution in immunodeficient mice using human fetal liver cells as a source of hematopoietic stem/progenitor cells [154]. The results of this study at variance with those observed in other studies with CD123 CARs using human adult bone marrow as a source of HSCs/HPCs is probably related to the higher expression of CD123 on fetal liver compared to bone marrow progenitors [154].

Potent CD19-directed CAR T cells immunotherapies have been used for the treatment of patients with relapsed/refractory B-ALL. However, CD-19-negative relapses are a major problem of this treatment, occurring in 30–40% of treated patients [155]. In an attempt to bypass this problem, Ruella and coworkers explored in preclinical models the possible strategy based on CD123 targeting [156]. Thus, these authors have shown that CD123 is highly expressed on leukemia-initiating cells and in primary CD19-negative patient leukemic blasts at diagnosis and at relapse after CAR T 19 administration [156]. Importantly, using an antigen-loss CD19-negative relapse xenograft model, it was provided evidence that CAR T 123, but not CAR T 19, recognized leukemic blasts, eradicated CD19-negative leukemia and prolonged survival of leukemic animals [156]. Furthermore, the combined use of CAR T 19 and CAR T 123 prevented antigen-loss relapses in xenograft models [156]. These observations support a strategy based on targeting CD19 and CD123 on leukemic blasts as a tool for both the treatment and the prevention of antigen-loss relapses after CD19-directed CAR T19 therapies. 

Given the consistent hematological toxicities observed in a preclinical model of human leukemia treated with CAR T 123, three different strategies for T cell termination have been explored: (i) Short-persisting messenger RNA-modified CD123-redirected CAR T cells (RIVA-CAR T 123); (ii) lentivirally transduced CD123-redirected CAR T cells (CAR T 123), subsequently depleted with the anti-CD52 monoclonal antibody Alenutuzumab and (iii) CAR T 123 co-expressing surface CD20 protein (CAR T 123-CD20), subsequently depleted with the anti-CD20 monoclonal antibody Rituximab [157]. Importantly, all these T-cell termination strategies maximized the therapeutic efficacy and overcome the potential toxicities for the normal hemopoietic system of AML CAR T 123 immunotherapy [157].

It is important to note that the potential toxicities of CAR T 123 immunotherapy are related to the recognition of low CD123-positive healthy tissues.

Another strategy to minimize the potential toxicity of CAR T 123 consisted in evaluating the effect of context-dependent variables capable of modulating CAR T cell functional profiles, such as CAR expression, CAR binding affinity and target antigen density [155]. Computational biology structural analysis allowed us to identify mutations in the anti-CD123 CAR antigen binding domain that can alter CAR binding affinity and CAR expression, without altering the general properties of these engineered T cells [155]. This strategy allowed the production of CD123 T cells with a good safety profile in terms of reduced capacity to affect normal tissues with low CD123 expression [158].

Recently, a new CAR T cell platform was developed using the expression of the toll receptor-like adaptor molecule, MYD88, and the tumor-necrosis factor family member CD40, tethered to the CAR molecule through a 2A linker system, providing a constitutive signal that drives CAR T proliferative, survival and anti-tumor signals to CD19^+^ and CD123^+^ hematological cancers [159]. However, the robust anti-tumor activity displayed by these CAR T cells was associated with induction of cachexia in animal models that requires specific strategies to reduce cytokine toxicity, such as administration of anti-TNFα antibody or selection of low cytokine producing T lymphocytes [159].

Other preclinical studies were based on the development of CAR T targeting both CD123 and CD33: The rationale of a CAR T expressing both anti-CD123 and anti-CD33 units consists in providing an optimal mechanism to target both bulk disease and leukemic stem cells [160]. The experimental studies carried out using CD123b-CD33bc CAR T cells exhibited a pronounced anti-leukemia activity, eliminating both the bulk and the leukemic stem cell population in primary AML samples [160]. Given the high potency of this double CAR T cell population, a safety-switch to protect against the potency of CAR T was developed [160].

Other studies reported the optimization of the procedures for the production of CAR T 123 involving human NK cells as a cellular target for anti-tumor therapy [161].

At this moment, CAR T 123-based immunotherapy is being investigated in 11 clinical trials for AML (Table 2). In 2015, Luio et al. reported a case report of a single relapsing AML patient treated with CAR T 123 cells, transduced using a fourth generation, apoptosis-inducible lentiviral CAR targeting CD123 [162]. The patient achieved a partial response and experienced an acute cytokine release syndrome, controlled by a single dose of Tolicizumab [162].

At the moment, the most consistent clinical experience was performed with the CAR T 123 reported by Mardiros et al. in 2013 [153] and developed as a clinical drug by the Mustang Bio Inc. and called MB-102. MB-102 is under evaluation in a phase I clinical trial in AML and BPDCN. Preliminary results on this ongoing clinical trial were presented at the 2017 ASH Meeting and at the America Association Carncer Research (AACR) Special Conference on Tumor Immunology and Immunotherapy (November 2018) [163,164]. To date, 18 patients have been enrolled and nine of them (seven with AML and two with BPDCN) have been treated. AML patients were heavily pretreated and all received allo-HSCT: Two patients were treated at 50 × 10^6^ dose of CAR T cells and one of them showed a response and received a second CAR123 T cell infusion and presented a decrease of leukemic blasts from 77% to about 1%; five patients received a higher dose level (200 × 10^6^) and two of them achieved a complete response and the remaining three had a stable disease [164,165]. The two BPDCN patients were treated at a dose of 100 × 10^6^ CAR123 T cells and one of them achieved a complete response. Importantly, in all patients the treatment was well tolerated, and patients showed only reversible and manageable toxicities, no patient displayed grade 3 or above cytokine release syndrome; finally, no treatment-related cytopenias were observed up to 12 weeks after the end of the treatment [164,165]. In 2018, the Food and Drug Administration has granted Orphan and Drug Designation to MB-102 for the treatment of BPDCN.

A trial on CAR T 123 was performed at the University of Pennsylvania using infusions of “bio-degradable” CAR T 123 cells (NCT2623852): These CAR T 123 cells were manufactured by electroporation of mRNA encoding the CAR and thus, at variance with CAR T cells transduced with lentivirus, have a limited capacity of expansion in vivo [166].

Thus, no significant anti-leukemic activity was observed in a clinical trial based on the use of these CAR T 123 cells; however, these cells were transiently detected in vivo and did not induce the release of cytokines [165]. This favorable safety profile supported the development of a phase-I trial of lentivirally-transduced second-generation CAR T 123 (CD123 CAR-41BB-CD3ζ); the treatment includes also a possible subsequent rescue allogeneic hematopoietic stem cell transplantation possibly related to induction of aplasia related to CD123 expression on hematopoietic stem/progenitor cells, with a conditioning regimen for the allogeneic HSCT involving the T-cell depleting agent Alemtuzumab to purge the CAR T 123 population in vivo.

The results of the various ongoing phase I clinical trials involving different CAR T 123 cell preparations will be essential to assess the real impact of this new CD123 targeting immunotherapy in the treatment of refractory/relapsing AML patients. The combination of allogeneic HSCT with CAR T cell therapy could represent the optimal strategy in the future for the treatment of relapsing/refractory AML patients.

Recent preclinical studies reported various strategies to improve the efficacy and to reduce the toxicity of CAR T cells. Mu and coworkers, to improve the efficacy and persistence in vivo of CAR T cells developed IL-15 expressing CAR123 T cells [154]. The results of these studies showed that CAR123 T cells expressing IL-15 resulted in an enhancement of effector anti-AML effect in vitro and in vivo [166].

A second study addressed the problem of controlling CAR123 T cell activity redirecting these cells using a switch-controllable universal CAR T platform (UniCAR) based on two main components: (a) A non-reactive, inducible second-generation CAR with CD28/CD3ζ stimulation for inert manipulation of T cells (UniCAR T) and (b) soluble targeting modules TM allowing UniCAR T reactivity in an antigen-specific manner [167]. A UniCAR T 123 displayed an efficient cytotoxic activity against patient-derived CD123^high^ leukemic cells [167]. Importantly, activation, cytolytic activity and cytokine release were strictly switch-controlled; furthermore, in contrast to conventional CAR T 123, UNICAR T 123 cells discriminate between CD123^high^ malignant leukemic cells and CD123^low^ healthy tissues [167].

## 3. Conclusions

The studies carried out in the last two decades support CD123 as a biomarker and therapeutic target for various hematological malignancies, and particularly for BPDCN and AML. CD123 targeting represents an effective therapeutic option for BPDCN patients and tagraxofusp, a drug based on a genetically engineered diphtheria toxin fused with IL-3, has shown robust activity and BPDCN was the first IL-3Rα targeting agent approved for the treatment of a hematological malignancy.

Anti-CD123 targeted therapies in pre-clinical studies and phase I-II clinical studies have supported the utility of IL-3-fused with diphtheria toxin, anti-CD123 neutralizing antibody-drugs, anti-CD123-targeting antibody-drug conjugate, CD123 × CD3 bispecific antibodies and dual-affinity retargeting therapies.

CD123 is also a diagnostic biomarker of MRD. MRD is an important prognostic factor for relapse and survival in AML and its eradication represents a key therapeutic strategy that may increase the number of AML patients with long-term survival [168]. At the moment, strategies that specifically target MRD are very limited [168]. Interestingly, CD123 represents a diagnostic biomarker of MRD and a potential therapeutic target for MRD eradication. Some trials are specifically evaluating the impact of CD123 targeting on MDR eradication prior to allogeneic stem cell transplantation [168].

Recent advances using CAR T-cells for B-cell ALL and non-Hodgkin lymphoma have generated a great interest for this strategy and for its extension to AMLs. CAR T cells targeting CD123 are being developed and are under evaluation in phase I/II clinical studies. The results of these initial studies will be of fundamental importance to assess the potentialities of this immunotherapy for AML patients and to plan future studies to define optimal combination of CAR T cell therapies with other current AML-therapies, particularly for refractory/relapsing AML patients.

## Figures and Tables

**Table 1 cancers-11-01358-t001:** European Leukemia NET (ELN)-risk stratification of molecular, genetic and cytogenetic acute myeloid leukemia (AML) alterations. Response rates were reported also, including CR (complete response), DFS (3-year, disease-free survival) and CR (3-year, overall survival).

Risk Category	Genetic Abnormality	Response to Therapy
Favorable	t(15;17)(q22;q12); PML-RARA	CR 81% DFS 46% OS 45%
	t(8;21)(q22;q22); RUNX1-RUNX1T1	
	inv(16)(q13.1q22) or t(16;16)(p13.1;q22) CBFB-MYH11	
	NPM1^mut^/FLT3^WT^ or FLT3-ITD^low = allelic ratio < 0.5^	
	Biallelic mutated CEBPA	
Intermediate-I	NPM1^mut^/FLT3-ITD^high = allelic ratio > 0.5^	CR 51% DFS 17% OS 18%
	NPM1^WT^/FLT3-ITD	
	NPM1^WT^/FLT3^WT^	
Intermediate-II	t(9;11)(p22;q23); MLLT3-KMT2A	CR 51% DFS 17% OS 18%
	Cytogenetic abnormalities not classified as favorable or adverse	
Adverse	t(6;9)(p23;q34.1); *DEK-NUYP214*	CR 37% DFS 2% OS 4%
	t(v;11q23.3): *KMT2A* rearranged	
	t(9;22)(q34.1;q11.2); *BCR-ABL1*	
	inv(3)(q21.3;q26.2) or t(3;3)(q21.3;q26.2); GATA2	
	MECOM (*EV11*)	
	−5 or del(5q); −7; −17/abn(17p)	
	Complex karyotype (three or more unrelated chromosomal abnormalities)	
	Monosomal karyotype	
	*NPM1^WT^/FLT3-ITD* ^high = allelic ratio > 0.5^	
	Mutated *RUNX1*	
	Mutated *ASXL1*	
	Mutated *TP53*	

**Table 2 cancers-11-01358-t002:** Therapeutic targeting using anti-CD123 monoclonal antibodies, interleukin-3 (IL-3) conjugated diphtheria toxin, CD3 × CD123 bispecific antibodies and dual-affinity re-targeting (DART) and chimeric antigen receptor (CAR) therapies related to CD123.

Drugs	Indication	Phase	Status	Identifiers
Tagraxofusp (human IL-3 conjugated to a truncated diphteria toxin)	AML or MDS	I/II	Completed	NCT00397579
Tagraxofusp	BPDCN, AMLK	I/II	Active, not recruiting	NCT02113982
Tagraxofusp	Relapsed/Refractory Multiple Myeloma	I/II	Ongoing	NCT02661022
Tagraxofusp	MDR-positive AML in remission	I/II	Ongoing	NCT02270463
Tagraxofusp	High-risk myeloproliferative neoplasms	I/II	Ongoing	NCTo2268253
Tagraxofusp	AML or high-risk MDS	I/II	Ongoing	NCT03113643
Talacotuzumab + Decitabine vs. Decitabine alone	AML ineligible to intensive chemotherapy	II/III	Ongoing	NCT02472145
Talacotuzumab	MDS or AML patients failing hypomethylating therapy	II	Terminated	NCT02992860
Talacotuzumab or Daratumumab	Low- or intermediate-risk MDS	II	Active, not recruiting	NCT0301134
IMGN632 (CD113-targeting antibody-drug conjugate, ADC)	Relapsed/refractory AML, BPDCN, ALL and other CD123^+^ hematological malignancies	I	Ongoing	NCT03386513
XmAb 14045 (anti-CD123/anti-CD3 bispecific monoclonal antibody)	Relapsed/refractory AML	I	Ongoing Partial clinical hold	NCT02730312
Flotetuzumab (MGD006) (Dual-affinity retargeting (DART) molecule targeting CD123 × CD3	Recurrent/refractory CD123-positive blood cancer	II	Not yet recruiting	NCT03739606
Flotetuzumab (MGD006, DART molecule targeting CD3 × CD123	Relapsed/refractory AML; Intermediate/high-risk MDS	I	Recruiting	NCT02152956
JNJ-63709178 (anti-CD123/anti-CD3 bispecific mAb)	Relapsed/refractory AML	I	Suspended	NCT02715011
antiCD123 CAR T (autologous lentivirally transduced) (CD123CAR-41BB-CD3)	Relapsed/refractory AML AlloHSCT is expected to be required in responding patients	I	Recruiting	NCT03766126
antiCD123 CAR T (autologous lentivirally transduced) (CD123CAR-CD28-CD3zeta-EGFRt)	Relapsed/refractory AML or relapsed BPDCN AlloHSCT is expected to be required in responding patients	I	Recruiting	NCT021159495
Universal (TCR KO) allogeneic antiCD123 CAR T (UCAR T123)	Relapsed/refractory AML or ELN adverse AML AlloHSCT is expected to be required in responding patients	I	Recruiting	NCT03190278
antiCD123 CAR T (allogeneic, donor-derived lentivirally transduced) (CD123CAR-CD28-CD3-EGFRt)	Relapsed AML after alloHSCT EGFR in CAR T construct allows for in vivo eradication of infused CAR T cells if needed with anti-EGFR mAb	I	Recruiting	NCT03114670
antiCD123 – antiCLL1 compound CAR T	Relapsed/refractory AML	I	Recruiting	NCT03631576
antiCD123 CAR T (autologous lentivirally transduced)	Relapsed/refractory AML (> 80% CD123^+^ leukemic blasts)	I	Recruiting	NCT03796390
antiCD123 CAR T (autologous or allogeneic lentivirally transduced)	Relapsed/refractory AML	I	Recruiting	NCT03556982
Multi-CAR T cell (autologous Muc1/CD33/CD38/CD56/CD123-specific T cells)	Relapsed/refractory AML	I	Recruiting	NCT03222674
4SCAR19 + 4SCAR123 (4th gen. CAR T targeting CD19 and CD123	B cell malignancies	I/II	Recruiting	NCT03291444

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
