# Peer review of "CD123 as a Therapeutic Target in the Treatment of Hematological Malignancies"

_cancers, 2019, doi:10.3390/cancers11091358_

Round 1

Reviewer 1 Report

Dr. Ugo Testa summarized the CD123 related targeting therapies, contains three major parts: fusion protein, specific antibodies, and CAR T cell, provided us comprehensive introductions on these therapies, which may benefit our readership in the area. Still, there are some minor issues needed to be addressed before considering publishing on our Journal, here are my points:

Authors should introduce the main types of hematological malignancies or some types related to CD123 since the title was “hematological malignancies” but authors mainly introduced AML. Line 29, “there are three different approaches” but authors introduced 5 in total. Line 272. “al” here? Line 274, “aloo” should be “allo”, consistency with the previous abbreviation. If possible, a diagram indicating CD123 domains will benefit our readership. Line 377, “CD1232” should be “CD123”. Authors spend a large paragraph to introduce the clone evolution in AML, please interpret the potential relationship between clone evolution and CD123 expression. The functions of CD123 in normal hematopoiesis need more details, if CD123 can serve a treatment target, the functions of CD123 in normal hematopoiesis should not be very critical, at least less important compared to malignancies, the potential side effects should be considered. Line 679, “only 0% among those with relapsed AML”, please double-check the “0%”, none of them? Line 716, “e3nriched”.

Author Response

1) Other hematological malignancies overexpressing CD123, such as various lymphoid malignancies, are now discussed.

2) Specific comments related to mistakes at the levelof lines 29, 272,274, 377, 679 and 716 have been corrected.

3) The functions of CD123 in normal hematopoiesis are now analyzed in more detail.

4) The potential relationship between AML clonal evolution and CD123 expression is now discussed.

Reviewer 2 Report

Review Report

cancers-584455 review article

CD123 as therapeutic target in the treatment of hematological malignancies; Testa, Pelosi and Castelli

Testa et al. have composed a comprehensive review on CD123 as therapeutic target in the treatment of hematological malignancies.  

The content of the article appears to be adequate for the description of the stated phenomena in both the quality and amount of data.

The comments are mostly English grammar corrections.

Specific comments

Lane 2: My suggestion for the title: CD123 as therapeutic target in the treatment of hematological malignancies

Lane 19: developed by Mustang

Lane 21: …, while a possible benefit...will have to be evaluated in the ongoing clinical studies.

Lane 45: Prognosis for AML patients >60 years has been improving but remains poor.

Lane 48: clonal hematopoiesis (CH)

Lane 51: delete “or not”

Lane 54: CHIP mutations

Lane 125: a molecular classification was proposed

Lane 167: delete “is”

Lane 208: IDH1

Lane 288: delete “that”

Lane 289: the new acquisition

Lane 320: the domains exert…

Lane 331: delete “is”

Lane 414: BPDCN represents the malignancy with the most promising clinical applications of CD123 targeted therapy.

Lane 419: … were reviewed recently (refs 71,72).

Lane 430: the median OS was 8-9 months

Lane 436: a genetically engineered fusion toxin was generated

Lane 609: the development of XY was reported which is composed of an antibody linked to a cytotoxic compound.

Lane 629: delete “showed”.

Lane 711: the development of XY was reported

Lane 716: enriched

Lane 820: meaning unclear: the only treatment with consistent efficacy (?)

Lane 826: AML patients were heavily pretreated and all received allo-HSCT.

Author Response

1) The title of the manuscript was modified as suggested by this reviewer.

2) All the mistakes have been corrected.

3) All the suggestions of specific corrections have been accepted.

Reviewer 3 Report

In this article, Testa et al discuss an emerging topic in treating hematological malignancies, which is targeting IL-3Ra, also known as CD123, mostly in acute myeloid leukemia (AML) and the rare disease entity blastic plasmacytoid dendritic neoplasm (BPDCN). They start by introducing AML including a detailed prognostic stratification based on cytogenetics and molecular aberrancies. They followed that by defining CD123 across the spectrum of hematological malignancies, followed by a comprehensive review of all available and pipeline drug classes that employ CD123 as a target, including ADC, BiTE, and CAR-T.

Overall, the manuscript is very well written, easy to read, and very informative to the hematology oncology audience in general, and to the hematological malignancy specialists in particular.

A few comments, if addressed, will further enhance the value of this review, as follows:

It will be reasonable to include in the abstract (lines 7-9) all types of hematological malignancies expressing CD123, basically all the ones discussed in the manuscript. At a first glance when you read the abstract, it seems the authors will only discuss AML, B-ALL, and BPDCN. In the first part of the introduction, the authors focus on AML and spend so much time discussing AML and cytogenetic/molecular subgroups, while they very lightly discuss the rest of hematological malignancies in which CD123 was used as target. It will be more reasonable to condense the AML paragraph, and change the approach to that opening paragraph from discussing AML, to brieflyintroducing all diseases in which CD123 is a target. At the end, the goal here is to educate about the value to CD123 as a target (which is well done by the authors), rather than discussing AML. Would add a line about hairy cell leukemia (HCL). Whether it has been tested in HCL or not, it will be very reasonable to discuss CD123 as a major surface marker in HCL. Similarly, what other NHL subtypes classically express CD123. Lines 73-80 contain the same info as in table 1. Consider deleting either/or. Authors also discussed ELN-risk stratifications only. If they decided to keep it, it will be more balanced to include NCCN as well and highlight the major discrepancies between the two networks. Authors might decide to remove both based on the way they see more fit to reshape that first paragraph as recommended in #2 above. In AML risk stratification: consider adding “other unknown” cyto abnormalities as intermediate risk. Line 48: preleukemic hematopoiesis (CH): please explain what is “CH”, it does not read well as the designated abbreviation for precedent. Line 64: t-AML, could also be secondary to h/o radiation therapy. Table 1: define complex karyotype (how many aberrancies to be complex). Authors uses IL-3Raand CD123 interchangeably across the manuscript. I suggest defining both at first use, then use CD123 solely CD123 reads much better compared to IL-3Ra. Line 377: typo for CD1232 instead of CD123. Lines 447-45: BPDCN is introduced for a second time. First introduction was in lines 424-419. Consider merging these two paragraphs, according to strategy in #2. Line 473: “tested in a few” might read better than “tested in few”. Line 716: typo for the third word “enriched”. When authors discuss the bispecific CD123 Ab, they seem to me as the same as bispecific T-cell engager (BiTE). I recommend using the appropriate designation for this specific drug class, as it will help the audience connecting these drugs to what is available and to what they use in the similar drug class (e.g. Blin…).

Author Response

1) Lymphoid malignancies, including hairy cell leukemia, Hodgkin lymphoma and acute lymphoblastic leukemia are now discussed as additional hematological malignancies displaying CD123 overexpression and are mentioned in the summary.

2) We prefere to leave a detailed analysis of AMLs in the introduction because we beleive that it is essential for a full understanding of the current studies of CD123 targeting.

3) Now in the manuscript it is used only CD123.

4) Lines 377 and 716: typo errors were corrected.

5) Line 64: it was now added that t-AMLs can be secondary also to radiation therapy.

6) Line 48: a correction was made to meet referee's request.

7) The NCCN risk stratification is briefly discussed.

8) BiTE (BI-specific T-cell engagers) are a class of bispecific monoclonal antibodies; they are fusion proteins consisting of two single-chain variable fragments (scFvs) of different antibodies on a single peptide: one of the two scFvs binds to T cells via CD3 receptor and the other to a tumor cell via a tumor cell via a specific tumor molecule. Not all the bispecific monoclonal antibodies targeting CD123 pertain to the BiTE group.